# Building towards Precision Oncology for Pancreatic Cancer: Real-World Challenges and Opportunities

**DOI:** 10.3390/genes11091098

**Published:** 2020-09-21

**Authors:** Yifan Wang, Anna Lakoma, George Zogopoulos

**Affiliations:** 1Department of Surgery, McGill University, Montreal, QC H4A 3J1, Canada; yifan.wang3@mail.mcgill.ca (Y.W.); anna.lakoma@mail.mcgill.ca (A.L.); 2Research Institute of the McGill University Health Centre, McGill University, Montreal, QC H4A 3J1, Canada; 3The Rosalind and Morris Goodman Cancer Research Centre, McGill University, Montreal, QC H3A 1A3, Canada

**Keywords:** pancreatic cancer, precision oncology, targeted therapy, biomarkers

## Abstract

The advent of next-generation sequencing (NGS) has provided unprecedented insight into the molecular complexity of pancreatic ductal adenocarcinoma (PDAC). This has led to the emergence of biomarker-driven treatment paradigms that challenge empiric treatment approaches. However, the growth of sequencing technologies is outpacing the development of the infrastructure required to implement precision oncology as routine clinical practice. Addressing these logistical barriers is imperative to maximize the clinical impact of molecular profiling initiatives. In this review, we examine the evolution of precision oncology in PDAC, spanning from germline testing for cancer susceptibility genes to multi-omic tumor profiling. Furthermore, we highlight real-world challenges to delivering precision oncology for PDAC, and propose strategies to improve the generation, interpretation, and clinical translation of molecular profiling data.

## 1. Introduction

Pancreatic cancer is projected to become the second-leading cause of cancer-related deaths in North America by 2030 [1]. Pancreatic ductal adenocarcinoma (PDAC) is the most common form of pancreatic cancer. Eighty percent of patients are diagnosed with advanced-stage PDAC, for which chemotherapy remains the only treatment with a survival benefit. The majority of patients who undergo curative-intent surgery ultimately develop metastatic recurrence [2]. This supports the notion that PDAC is a systemic disease and underscores the importance of chemotherapy across all stages. Two phase III trials have established FOLFIRINOX (fluorouracil, leucovorin, irinotecan, oxaliplatin) and gemcitabine plus nab-paclitaxel as first-line chemotherapies for metastatic PDAC [3,4]. However, these regimens remain associated with a median survival of less than one year, and importantly, neither is informed by biomarker status. Table 1 and Table 2 summarize landmark clinical trials evaluating systemic therapy for PDAC in the palliative and adjuvant settings [3,4,5,6,7,8,9,10,11].

Over the past decade, advances in next-generation sequencing (NGS) have revealed the molecular heterogeneity of PDAC. This heterogeneity may partly explain the limited effectiveness of “one-size-fits-all” chemotherapy. Distinct molecular subtypes of PDAC have been identified, and fall broadly into two categories: (1) PDAC driven by germline predisposition and (2) PDAC driven by somatic oncogenic alterations. Despite emerging evidence showing subtype-specific prognosis and therapeutic sensitivity, biomarker-driven treatment stratification is currently not routine practice for PDAC. In fact, clinicians continue to treat PDAC as a single disease with only patient performance and comorbidities dictating therapy selection. There is an urgent need to bridge the gap between the expanding capacity for NGS and the limited infrastructure in place to integrate these technologies into clinical practice.

## 2. Inherited Predisposition to PDAC

It has long been recognized that PDAC runs in certain families, giving rise to the concept of familial pancreatic cancer (FPC) [12]. Prospective epidemiological studies of FPC families have shown that an individual’s PDAC risk is correlated with the number of affected family members [13]. Furthermore, individuals from FPC kindred were also found to have higher rates of breast and ovarian cancer. These observations provided motivation to investigate germline mutations associated with PDAC predisposition, initially through targeted sequencing of known cancer predisposition genes, and subsequently, through whole-exome and whole-genome sequencing approaches.

Together, these studies have revealed that 3–8% of PDAC patients harbor an underlying deleterious germline mutation in a cancer predisposition gene [14,15,16,17]. Moreover, this proportion is higher in patients that have a family history of PDAC, as well as those from founder populations (e.g., Ashkenazi Jewish, French Canadian) [18,19]. In the following sections, we will discuss PDACs arising from germline mutations in specific DNA repair pathway genes, focusing on their molecular hallmarks and therapeutic implications. The distinct biology of PDACs associated with germline *ATM* mutations is outside the scope of this paper, but is reviewed in detail by Nanda et al. in this Special Issue [20].

### 2.1. Homologous Recombination Repair Deficient PDAC

The homologous recombination repair (HRR) pathway maintains genomic integrity by repairing double-strand DNA breaks [21]. Pathogenic germline mutations in HRR genes, such as *BRCA1*, *BRCA2*, and *PALB2*, account for the largest known fraction of inherited PDAC susceptibility. PDACs that have HRR deficiency (HRRd) may be targeted therapeutically using drugs that induce DNA damage. Platinums cause double-strand DNA breaks that cannot be effectively repaired by HRRd cells [22]. Poly(ADP-ribose) polymerase (PARP) inhibitors prevent the repair of single-strand DNA breaks through both catalytic inhibition of PARP and PARP trapping, resulting in the accumulation of irreparable DNA damage [22,23,24].

We and others have shown the in vitro and in vivo sensitivity of HRRd PDAC to platinums and PARP inhibitors (PARPi) [25,26,27]. Retrospective studies have demonstrated that patients with HRRd PDAC have a survival benefit when treated with platinum-based therapy [19,28]. The landmark POLO trial evaluated maintenance olaparib (PARPi) in patients with platinum-sensitive metastatic PDAC and a germline *BRCA1* or *BRCA2* mutation, and showed longer progression-free survival with olaparib compared to placebo (7.4 vs. 3.8 months) [29]. Recently, a phase II trial showed a 65% response rate with first-line gemcitabine-cisplatin in patients with advanced-stage PDAC and a germline *BRCA1*, *BRCA2,* or *PALB2* mutation [30]. The addition of veliparib (PARPi) to gemcitabine-cisplatin resulted in similar response rates, progression-free survival, and overall survival compared to gemcitabine-cisplatin alone. Thus, these data do not support a synergistic role of veliparib with platinum-based therapies. Albeit, it is possible that other PARPis, particularly later generation PARPis such as talazoparib, may have better efficacy in combination with platinums.

Despite these encouraging results, one recurring observation across these trials is that a subset of patients does not respond to platinum or PARPi, despite harboring a germline HRR gene mutation. In keeping with Knudson’s two-hit hypothesis, PDACs that are driven by a germline pathogenic mutation should exhibit inactivation of the second wildtype allele [31]. In fact, biallelic, but not monoallelic inactivation of HRR genes leads to distinct genome-wide hallmarks, including COSMIC single base substitution signature 3, rearrangement signature 5, and deletions with microhomology [32]. HRDetect is a composite model that integrates these characteristic mutational hallmarks to predict HRR deficiency [33]. In some cases, biallelic loss of HRR function may also be caused by *BRCA1* promoter hypermethylation or biallelic somatic mutations.

In the absence of a second hit, the germline HRR gene mutation is considered a passenger event that is not etiologically implicated in cancer development (i.e., occurring by chance, at population allele frequency). Importantly, these monoallelically inactivated PDACs are characterized by a low HRDetect score and reduced sensitivity to first-line platinum therapy compared to PDACs with biallelic inactivation [26,34]. The “etiologic index” is a metric that estimates the relative risk that a germline mutation in a tumor suppressor gene is etiologically implicated in the development of a given cancer as opposed to a passenger event [35]. In contrast to breast and ovarian cancers, the etiologic index of *BRCA2* is higher compared to *BRCA1* in PDAC (4.8 vs. 1.7). This parallels the higher PDAC risk in individuals carrying a germline *BRCA2* mutation compared to those with a germline *BRCA1* mutation [36]. Together, these findings highlight the limitations of stratifying treatment based on germline HRR gene mutational status alone and may partly explain the heterogeneous responses seen in published trials. This provides motivation to evaluate combined germline and somatic HRR biomarker assays in future clinical trial design.

### 2.2. Mismatch Repair Deficient PDAC

The DNA mismatch repair (MMR) pathway functions to identify and correct mismatched DNA base pairs. Lynch syndrome (LS) is an autosomal dominant condition caused by pathogenic germline mutations in the MMR genes (*MLH1*, *MSH2*, *MSH6*, *PMS2*). Individuals with LS have an approximately 9-fold increased risk of developing PDAC [37]. In the absence of LS, MMR deficiency can arise sporadically via somatic MMR gene mutations or epigenetic alterations, such as *MLH1* promoter hypermethylation.

PDAC has typically been considered a weakly immunogenic tumor and a poor candidate for immunotherapy [38]. Although only 1–2% of incident PDAC cases are MMR-deficient (MMRd), there is considerable interest due to the biological rationale for immunotherapy in this subtype. Loss of MMR function results in an elevated tumor mutational burden (TMB) and microsatellite instability (MSI-H) [39]. Expression of nonsynonymous mutations can produce tumor-associated neoantigens that elicit intratumoral CD8+ T-cell infiltration [40]. This antitumor immune response is usually counterbalanced by inhibitory checkpoints, such as programmed cell death-1 (PD-1). Because of their increased immunogenicity, MMRd tumors are susceptible to immune checkpoint inhibitors (ICI) that reactivate CD8+ T-cell mediated cytotoxicity.

In a phase II trial by Le et al., MMRd/MSI-H PDACs (*n* = 8) showed a 62% objective response rate with pembrolizumab (anti-PD-1 monoclonal antibody) [41]. Similarly, Hu et al. showed a 57% response rate in MMRd/MSI-H PDACs treated with pembrolizumab [42]. However, in the KEYNOTE-158 phase II trial of non-intestinal MMRd/MSI-H cancers, only 18% of PDACs (*n* = 22) showed an objective response, which was the lowest rate among the investigated cancers [43]. These findings reveal tissue-specific differences in ICI sensitivity, and highlight the need for additional biomarkers to identify patients that are most likely to benefit from ICI. Recently, the Food and Drug Administration (FDA) approved pembrolizumab in second-line for advanced solid tumors with a TMB above 10 mutations/megabase. Although response rates are generally higher in tumors with higher TMB, it is noteworthy that 6.7% of patients who had an objective response in the KEYNOTE-158 trial had a TMB below 10. By corollary, a TMB-based biomarker may provide an opportunity to evaluate the efficacy of ICI in HRRd PDAC, which typically has a higher TMB compared to sporadic PDAC [39].

## 3. Somatic Alterations in PDAC: Drugging “Undruggable” Drivers

In the previous section, we considered how the discovery of hereditary germline mutations provided an initial framework for the practice of cancer genetics in PDAC. However, most PDACs do not arise in the context of germline predisposition. In recent years, multi-omic tumor profiling has uncovered the molecular heterogeneity of PDAC. In the next section, we will review the common PDAC driver genes and transcriptomic subtypes, with an emphasis on their prognostic and therapeutic relevance.

### 3.1. KRAS

Oncogenic *KRAS* mutations are the most frequent drivers in PDAC, and are present in 90–95% of cases [44]. These mutations reduce the ability of the KRAS GTPase to hydrolyze GTP, leaving it in a constitutively active GTP-bound form that promotes oncogenic downstream signaling. Considering its prevalence, mutant KRAS is a compelling therapeutic target in PDAC. To date, direct KRAS blockade has proven challenging, because of the lack of suitable binding pockets for small molecule inhibitors [45]. Moreover, inhibition of the downstream RAF-MEK-ERK cascade is typically ineffective due to compensatory activation of negative feedback loops [46].

The first-in-class KRAS inhibitor is AMG510, a small molecule inhibitor that specifically targets KRAS G12C [47]. AMG510 forms a covalent bond with GDP-bound KRAS G12C and inhibits GTP binding. The phase I CodeBreak 100 trial showed a 100% disease control rate in *KRAS* G12C-mutant advanced non-small cell lung cancer [48]. This study was subsequently expanded to other solid tumors, but showed only modest efficacy. Of 8 evaluable PDAC cases, 6 achieved stable disease as best response, whereas the remaining 2 progressed [49]. In addition, only 2–3% of PDACs are driven by a *KRAS* G12C mutation, which limits its overall applicability in PDAC.

Recently, a SOS1::KRAS inhibitor (BI-3406) has been developed, which binds to the catalytic domain of SOS1 and blocks the exchange of GDP for GTP on KRAS, thereby inhibiting KRAS activation [50]. Importantly, BI-3406 also attenuates feedback reactivation induced by MEK inhibitors. In KRAS-driven patient-derived xenografts, dual BI-3406 and MEK inhibition (trametinib) achieved substantial tumor regression, with the combination more effective than either drug alone. In contrast to covalent *KRAS* G12C-specific inhibitors, BI-3406 is active across the majority of mutant *KRAS* alleles, including the G12D and G12V mutants that predominate in PDAC. There is an ongoing phase I trial (NCT04111458) evaluating the clinical benefit of SOS1: KRAS inhibition, alone and in combination with trametinib, in advanced *KRAS*-mutated cancers.

The minority of PDACs that harbor wildtype *KRAS* is of particular interest. In the absence of oncogenic KRAS signaling, PDACs tend to exhibit alterations in other RAS pathway genes or alternate oncogenic drivers [44]. Importantly, several of these alternate drivers are potentially targetable with existing therapies. For example, canonical *BRAF* mutations may be sensitive to RAF/MEK inhibition, whereas *ERBB2* amplification may respond to pertuzumab/trastuzumab [51,52]. Furthermore, *KRAS*-wildtype PDACs are also enriched for oncogenic gene fusions, including *NRG1, RET,* or *NTRK* (discussed below), which may be targetable using tyrosine kinase inhibitors. These findings provide rationale to investigate alternate oncogenic drivers and gene fusions in *KRAS*-wildtype PDACs, if not done simultaneously.

### 3.2. Tumor Suppressor Genes

The other common PDAC driver mutations implicate tumor suppressor genes: *TP53*, *CDKN2A*, and *SMAD4*.

*TP53* is mutated in 60–80% of PDACs [53]. Functionally, p53 maintains genomic integrity by mediating cell cycle arrest and apoptosis in response to stress conditions, such as DNA damage and oncogene activation [54]. Inactivation of p53 allows oncogene-expressing cells to proliferate unabated. There has been interest in compounds that can restore p53 tumor suppressor function. APR-246 is a first-in-class small molecule that selectively induces apoptosis in *TP53*-mutant cells. APR-246 combined with azacitidine showed a 75% response rate in *TP53*-mutant myelodysplastic syndromes and acute myeloid leukemia [55]. To date, this agent has not been investigated in PDAC.

*CDKN2A* inactivation is present in 30–50% of PDAC cases [56]. *CDKN2A* exerts its tumor suppressor role by inhibiting cyclin-dependent kinases (CDK) 4/6 from initiating G1/S cell cycle progression. CDK4/6 inhibitors are approved for advanced estrogen receptor-positive breast cancer, where they prevent hormone-dependent cell cycle entry [57]. However, CDK4/6 inhibitor monotherapy has shown only modest responses in other tumor types, suggesting that combination therapy is likely required [58]. Since CDK4/6 inhibitors result in G1 arrest, they may antagonize the effects of cytotoxic chemotherapies that act on cells during the synthesis and mitosis phases. To this end, Salvador-Barbero et al. showed that the CDK4/6 inhibitor was more effective in suppressing the growth of PDAC cells when administered after, rather than before, cytotoxic chemotherapy [59]. These data highlight the importance of judicious sequencing of CDK4/6 inhibitors and chemotherapy in future clinical trial design.

*SMAD4* mediates tumor suppression through the TGF-β signaling pathway, and is inactivated in 30–40% of PDACs [56]. TGF-β blockade has demonstrated promising antitumor activity in preclinical models [60,61]. To date, however, early-phase clinical trials evaluating TGF-β inhibitors (e.g., galunisertib) have shown only modest efficacy in PDAC [62]. In this Special Issue, Hsieh et al. leveraged publicly available datasets to show that *SMAD4* deletion is associated with shorter disease-free survival in PDAC [63]. They also showed that *SMAD4*-deleted PDAC cells have increased sensitivity to cell cycle-targeting drugs due to upregulation of cell cycle-related genes.

### 3.3. NTRK Gene Fusions

Fusions involving the neurotrophic receptor tyrosine kinase (*NTRK*) gene can function as oncogenic drivers. Although *NTRK* fusions are present in only 1% of PDACs, this subtype is of interest due to the emergence of small molecule tyrosine kinase inhibitors (entrectinib, larotrectinib) [64]. Pishvaian et al. reported 3 patients with gene fusion-positive PDAC (*NTRK*, *n* = 2; *ROS1*, *n* = 1), all of whom had a sustained partial response to entrectinib [65]. Recently, an updated analysis of 3 basket trials (STARTRK-2, STARTRK-1, and ALKA-372-001) included 3 PDAC patients treated with entrectinib who had a median duration of response of 10 months [66]. Entrectinib and larotrectinib have been FDA-approved for advanced *NTRK* fusion-positive solid tumors that have progressed following prior therapies. The approval also extends to first-line use when there are no effective alternative therapies. Recently, Demols et al. reported using an anti-pan-TRK immunohistochemical (IHC) assay as an initial screen for *NTRK* fusion-positive biliopancreatic cancers [67]. Positive staining by IHC triggered confirmatory testing using an RNA-based NGS panel. Adopting a two-step diagnostic approach, with an initial IHC assay, followed by NGS confirmation of IHC-positive cases, may be a resource-effective strategy to identify rare, but clinically significant, *NTRK* fusions. Interestingly, *NTRK* fusion-positive PDACs appear to be mutually exclusive with oncogenic *KRAS* mutations, further underscoring the need to screen for alternative drivers in the minority of PDACs with wildtype *KRAS* [65].

### 3.4. Transcriptomic Subtypes

Multiple groups have proposed transcriptomic classifications of PDAC. Collisson et al. originally analyzed gene expression microarrays and identified 3 subtypes: classical, quasi-mesenchymal, and exocrine-like [68]. Subsequently, Moffitt et al. used gene expression microarray data and applied a virtual microdissection algorithm to digitally separate tumor from stromal gene expression [69]. They defined 2 tumor-specific subtypes: classical and basal-like. Bailey et al. performed RNAseq-based analysis, and defined 4 subtypes: pancreatic progenitor, squamous, aberrantly differentiated endocrine-exocrine (ADEX), and immunogenic [70]. Their pancreatic progenitor subtype showed strong overlap with both the Collisson and Moffitt classical subtypes, whereas their squamous subtype closely resembled the Collisson quasi-mesenchymal and Moffitt basal-like subtypes. The Cancer Genome Atlas (TCGA) consortium applied the clustering techniques from the Collisson, Moffitt, and Bailey studies, and corroborated these overlaps [71].

Synthesizing these studies is challenging because of sample and computational variability, but two tumor-specific subtypes have consistently emerged across all studies: classical and basal-like. Biologically, the classical subtype is characterized by high expression of epithelial genes, whereas the basal-like subtype exhibits upregulation of mesenchymal genes. Interestingly, molecularly similar subtypes have been described in breast and bladder cancers, supporting the existence of an underlying biological distinction [69]. Importantly, the basal-like PDAC subtype is associated with poor differentiation, and has shown a worse overall survival compared to classical PDAC across all studies [68,69,70,72]. In the ESPAC-3 trial, basal-like PDACs had a worse response to adjuvant 5-fluorouracil/leucovorin [73]. Similarly, in the COMPASS trial, basal-like PDACs treated with modified FOLFIRINOX showed the worst progression-free survival (shorter than basal-like PDACs treated with gemcitabine plus nab-paclitaxel) [72]. These data suggest that gemcitabine-based cytotoxic therapies should be favored over FOLFIRINOX in basal-like PDACs, especially considering the toxicity associated with FOLFIRINOX.

Although these recent studies suggest that transcriptomic subtyping is clinically relevant, whole transcriptome sequencing is resource-intensive. To this end, IHC classifiers have been developed as surrogate assays. High GATA6 expression has been identified as a marker for the classical subtype [72,74]. Recently, an IHC ratio combining GATA6 with a basal-like marker, KRT17, has shown improved ability to discriminate between transcriptomic subtypes compared to GATA6 alone [26]. One caveat to recognize is that the classical vs. basal-like subtypes are non-dichotomous, but rather, exist within a transcriptional continuum. In fact, single cell analyses have revealed coexistence of classical and basal-like subpopulations within the same tumor [75]. Understanding the biological nuances of PDACs that lie in the middle of this spectrum will be important to rationalize subtype-driven therapies.

### 3.5. Epigenetic Alterations in PDAC

Epigenetic dysregulation is frequently observed in PDAC [76]. In fact, histone deacetylases (HDAC) and DNA methyltransferases (DNMT) are overexpressed in up to 80% of PDACs, and promote cell cycle progression and proliferation [77,78]. DNMT inhibitors, such as azacitidine and decitabine, have shown efficacy in PDAC xenografts [79,80]. There are ongoing early-phase clinical trials evaluating DNMT inhibitors in combination with gemcitabine in advanced PDAC. Recently, Nicolle et al. performed an integrative multi-omic analysis using patient-derived xenografts (PDX), and suggested that PDAC transcriptomic subtypes are driven by distinct DNA methylation patterns [81]. This observation is functionally important considering the reversible nature of epigenetic alterations, which makes them a compelling therapeutic target. To this end, Lomberk et al. showed that the MET oncogene modulates basal-specific super-enhancers, and that MET inhibition leads to the transition from a basal-like to a classical PDAC phenotype [78].

Since epigenetic aberrations are often early events in carcinogenesis, identification of DNA methylation biomarkers may facilitate early detection of PDAC [82,83]. Eissa et al. showed that applying a two-gene methylation panel (*BNC1*, *ADAMTS1*) to peripheral cell-free DNA could detect PDAC with high sensitivity and specificity [84]. Thus, DNA methylation-based biomarkers may have a value as non-invasive tests for early detection of PDAC [85].

## 4. Real-World Challenges in Implementing a PDAC Precision Oncology Program

The rapid evolution of NGS technologies and the increasing number of biomarker-driven therapies signal a new dawn for PDAC treatment paradigms. However, a variety of operational challenges makes it difficult to implement PDAC precision oncology programs into clinical practice. In the following sections, we will address practical hurdles that exist in (1) performing molecular testing, (2) interpreting results of molecular testing, and (3) accessing molecular-driven therapies.

### 4.1. Performing Germline Testing and Tumor Molecular Profiling

To date, tumor molecular profiling has been essentially undertaken in a research setting, using whole-genome and whole-transcriptome sequencing approaches that are primarily aimed at discovery. This multi-omic approach is cost- and resource-intensive, and its use is difficult to justify outside of a research protocol. Recently, companion diagnostic tests, such as MSK-IMPACT and FoundationOne CdX, have been developed that screen a panel of several hundred cancer-related genes [86,87]. These gene panels are curated to include common mutations that have potential targeted treatment options. Is PDAC molecular profiling using a companion multi-gene panel a research or clinical test?

The National Comprehensive Cancer Network (NCCN) guidelines recommend considering somatic testing for patients with metastatic PDAC [88]. Recently, the European Society of Medical Oncology (ESMO) issued guidelines for the use of NGS in various cancers [89]. Routine multi-gene NGS testing was recommended for non-small cell lung cancer, cholangiocarcinoma, prostate cancer, and ovarian cancer on the basis of level I evidence. For PDAC, ESMO recognized the clinical benefit of matched therapies for germline *BRCA1*/*BRCA2*-mutated, MSI-H, and *NTRK* fusion-positive cancers. However, evidence supporting the benefit of targeted therapy for PDACs with somatic *BRCA1*/*BRCA2* mutations was deemed insufficient. Thus, ESMO did not recommend routine tumor (somatic) NGS testing for PDAC, but urged centers to propose multi-gene sequencing in the context of molecular profiling programs to increase access to innovative drugs. One argument against comprehensive tumor profiling may be that multi-gene panels screen a number of genes that may have limited relevance to PDAC. However, iterative testing using standalone single-gene tests is both time-consuming and wastes precious tissue, all at comparable cost to simultaneous multi-gene sequencing.

The distinction between a research or clinical test has practical ramifications. On a research basis, each patient needs to be enrolled on a study protocol by a dedicated coordinator. Such a model creates human resource and financial bottlenecks that limit scalability at an institutional level. In contrast, clinical tests do not face the same recruitment challenges, especially if the consent process for tumor molecular profiling and germline testing is embedded within clinical care pathways. PDAC precision oncology programs currently fall in a grey zone between research and routine practice, but are primed to coexist with clinical care. In the current climate, there is room for oncologists and surgeons to discuss the value of multi-gene testing with PDAC patients at diagnosis in the ambulatory clinic setting, especially if institutional NGS programs are introduced.

One important practical consideration is how to streamline the acquisition of PDAC tissues for molecular profiling. To prevent repetitive procedures, it is crucial that molecular profiling can piggyback on tissues that are obtained as part of routine clinical care (i.e., diagnostic biopsies or surgical resection specimens). This is especially important for PDAC patients, as additional biopsies for the purpose of tumor molecular profiling may incur added risk and delay their start of chemotherapy. Whole genome and whole transcriptome sequencing generally require fresh tissues to be snap-frozen [90]. The extent of human resources required for real-time tissue processing limits its feasibility outside of a research protocol. Thus, molecular profiling tests should be optimized for formalin-fixed paraffin-embedded (FFPE) tissues that can be processed following standard, less time-sensitive protocols.

Diagnostic procedures should be planned while being mindful of sample considerations for tumor molecular profiling. PDAC is typically characterized by low tumor cellularity and a strong desmoplastic reaction. At a tumor cellularity below 10–20%, the analytical sensitivity of NGS platforms is insufficient to detect low-frequency variants, resulting in increased risk of false-negative results [91,92]. To overcome these limitations, endoscopic ultrasound with fine-needle biopsy, which yields larger tissue cores with preserved architecture, may be preferable compared to fine-needle aspiration. Solid organ metastases can have a reduced desmoplastic reaction compared to the primary tumor, and may represent better targets for sampling [93]. Integration of tumor enrichment strategies, such as macrodissection or laser-capture microdissection, will be important to enable molecular profiling of low-cellularity specimens.

To date, the phase III POLO trial provides the strongest evidence for a biomarker-driven treatment approach in PDAC. Considering the importance of identifying germline HRR gene mutations, germline testing is now recommended for all patients diagnosed with PDAC [88]. However, implementing genetic counselling (GC) and germline testing is challenging within traditional care delivery models. Patients with PDAC have a limited life expectancy, competing demands for medical care, and evolving priorities. There is a short window of opportunity for GC and germline testing, but wait-times to access clinical GC services are currently prohibitive [94]. Yurgelun et al. showed that an automatic referral system with same-day GC consultation resulted in higher uptake of GC and germline testing compared to the traditional system that relied on active oncologist referral (41 vs. 17%) [95]. We and others have shown that a point-of-care GC strategy, wherein genetic counsellors are embedded within the ambulatory oncology clinic, results in high levels of patient participation and satisfaction [96,97]. In addition to streamlining existing referral pathways, innovative GC practice models are needed to meet the increasing demands being placed on already overstretched clinical GC departments. To this end, the Dutch DNA-direct model replaces initial face-to-face GC by pre-test telephone appointments supplemented with access to written and online educational materials [98]. At the Royal Marsden Hospital, non-geneticists (surgeon, oncologist, specialized oncology nurse) who are certified following completion of an online training module can deliver pre-test counselling and consent patients to germline testing at the point of care [99]. A post-test appointment with a clinical geneticist is scheduled for any patient that is found to carry a pathogenic germline mutation, or upon patient request. Transitioning towards modern, patient-centered GC models may allow more sustainable delivery of genetic testing for PDAC patients. This approach may also facilitate cascade testing and initiation of surveillance strategies for at-risk relatives.

### 4.2. Returning and Interpreting Molecular Profiling Results

The expanding number of molecular alterations identified in PDAC is magnifying the challenge of clinical interpretation. The actionability of a given variant exists within an evolving spectrum of clinical and preclinical evidence. To facilitate interpretation of these data, a molecular tumor board (MTB) could be incorporated into the framework of existing multidisciplinary cancer conferences. A MTB would ideally bring together clinicians, medical geneticists, molecular pathologists, genetic counsellors, bioinformaticians, and research scientists [100]. The MTB examines the patient’s clinical, pathology, and molecular results, and formulates recommendations for targeted treatment opportunities. Implementation of a structured MTB may increase physicians’ willingness to perform molecular profiling [101]. In addition, virtual MTBs done in collaboration with partner institutions can provide a forum that leverages the clinical and bioinformatic expertise and experience of multiple groups. This setup also provides an opportunity for community physicians to participate in clinical care discussions [102]. Furthermore, variant interpretation can be enhanced with cognitive computing platforms such as IBM’s Watson for Genomics (WfG). Studies have shown that WfG can identify additional potentially actionable variants that were not prioritized by manual curation alone [103]. These findings demonstrate the value of leveraging artificial intelligence platforms for interpretation of molecular data.

Tumor-based molecular profiling may reveal incidental germline variants that are clinically significant in up to 12% of patients [104,105]. This can pose significant challenges for treating physicians who may not be prepared to manage incidental findings (may also be referred to as secondary findings) that can have broader implications for the patient’s relatives. It is important that centers offering tumor-based NGS testing provide appropriate pre-test counselling to inform patients regarding the possibility of identifying hereditary cancer susceptibility. In addition, patient preferences regarding confirmatory testing of potential germline variants and return of incidental germline results should be solicited. Several studies have shown that, even in the advanced cancer setting, the majority of patients wish to be informed of such incidental results [104]. For these patients, a post-test consultation with a genetic counsellor should be streamlined to organize confirmatory germline testing, to counsel the patient regarding the associated risks for themselves and their relatives, and to initiate cascade testing.

With the increasing amount of NGS testing performed, there is a pressing need to better communicate and share these data, both locally and at a broader level. Molecular profiling data have traditionally been generated, analyzed, and stored in silos, whether at an institutional level or within specific clinical departments. The lack of standardization in mutation-calling pipelines, variant annotation formats, and use of clinical ontologies results in fragmented knowledge bases that are difficult to aggregate. To this end, the Global Alliance for Genomics and Health (GA4GH) is an international alliance that is building frameworks and defining technical standards to enable the responsible sharing of genomic data [106]. At a local level, the molecular profiling report should be integrated in the patient’s electronic medical record and highlight the most critical information in a concise manner. As these records may infer germline information, security measures should be applied that are similar to those used to protect germline testing results from medical genetics consultations. An expert consensus working group led by the Association for Molecular Pathology has outlined recommendations for a standardized reporting framework [107]. These recommendations stress the importance of using standardized nomenclature and prioritizing clinically relevant variants using a tier-based system. Continued efforts to harmonize NGS data will be important to maximize the clinical impact of molecular profiling initiatives.

### 4.3. Validating and Accessing Biomarker-Driven Therapies

One consistent finding across all PDAC tumor-profiling studies is that only a small fraction of eligible patients received biomarker-driven therapy. Through the MSK-IMPACT panel, 26% of PDAC patients were found to have a potentially actionable mutation, but only 1% received matched therapy [108]. Similarly, the COMPASS study identified actionable mutations in 30% of patients, but only 8% received biomarker-driven therapy [72]. Recently, the Know Your Tumor registry identified actionable molecular alterations in 282 (26%) of 1082 PDAC patients [109]. Of these, patients who received molecularly matched therapy in second- or further-line had significantly longer median overall survival compared to patients who only received unmatched therapy. However, it is noteworthy that 76% of patients with actionable findings did not receive matched therapy. Together, these studies highlight that attempts to match patients with biomarker-driven therapies too often fall short.

For PDAC patients with molecular alterations that have FDA-approved therapies (i.e., germline *BRCA1*/*BRCA2* mutation, MSI-H, *NTRK* gene fusion), matched therapy is more readily accessible. For the remainder of patients, clinical trials represent the best opportunity to access off-label targeted therapy. To overcome the limitations of traditional clinical trial design, several innovative protocols have been developed over the past decade, which allow the investigation of multiple target-treatment pairs in parallel. Basket trials (e.g., NCI-MATCH, NCT02465060; CAPTUR, NCT03297606) evaluate the effect of one drug on patients with a specific molecular aberration, independent of tumor histology [110,111]. In umbrella trials (e.g., PRIMUS-004, EudraCT 2018-003971-37), patients with the same tumor type are assigned to one of several sub-arms investigating a biomarker-driven targeted therapy [112]. Recently, adaptive trial designs have emerged, which provide opportunities to modify specific trial elements (e.g., treatment arms, randomization ratios, etc.) on the basis of interim efficacy data analysis. In this manner, adaptive designs can reduce the cost and duration of drug development, while sparing patients from receiving ineffective treatments. Importantly, these trial designs require close collaboration between academic, industry, and regulatory partners.

Despite the increasing availability of biomarker-driven clinical trials, real-world patient enrollment is resource-intensive [113]. Matching patients to clinical trials requires a detailed knowledge of both patient characteristics and trial eligibility criteria. Undertaking this task manually is time-consuming and can often deter clinicians from enrolling eligible patients at the point-of-care. To this end, several trial-matching systems have been developed, which leverage artificial intelligence and natural language processing tools to facilitate identification of eligible patients for relevant clinical trials. For example, IBM’s Watson for Clinical Trial Matching (CTM) is a software platform that analyzes data from a patient’s electronic medical record against protocol eligibility criteria. In a pilot study conducted at the Mayo Clinic, implementation of the Watson for CTM system was associated with an increase in clinical trial enrollment and reduced screening times [114]. Thus, implementing automated clinical trial matching algorithms in the ambulatory oncology setting may supplement clinician review and increase clinical trial participation.

In the previous section, we have outlined several strategies to increase clinical trial accrual. However, it is not always possible to sufficiently power biomarker-driven clinical trials that involve rare molecular alterations. In this context, preclinical models can be leveraged in parallel to investigate the efficacy of potential therapies. As reviewed by Pereira et al. in this Special Issue, patient-derived xenograft (PDX) and patient-derived organoid (PDO) models closely recapitulate the mutational spectrum of the parent tumor [115]. Tiriac et al. showed that PDOs can be generated with a 75% efficiency from fresh PDAC samples, and that their therapeutic sensitivities paralleled patient outcomes [116]. In this manner, integration of preclinical models can provide a scalable platform to test and contribute to the validation of potential targeted therapies. Together with clinical trial evidence, this may accelerate access to targeted treatments for patients with PDAC.

## 5. Conclusions

The advent of NGS technologies has provided unprecedented insight into the biology of PDAC. The ever-increasing number of potentially actionable alterations and biomarker-driven therapies in PDAC provide considerable reason for optimism. However, there are numerous operational challenges that currently limit our capacity to integrate precision oncology in routine clinical practice. Figure 1 depicts a strategic framework to address these challenges. Foresight and investment from all stakeholders will be essential to develop the clinical, technological, and regulatory infrastructures needed to validate precision oncology approaches and translate NGS technologies into patient impacts.

## Figures and Tables

**Figure 1 genes-11-01098-f001:**
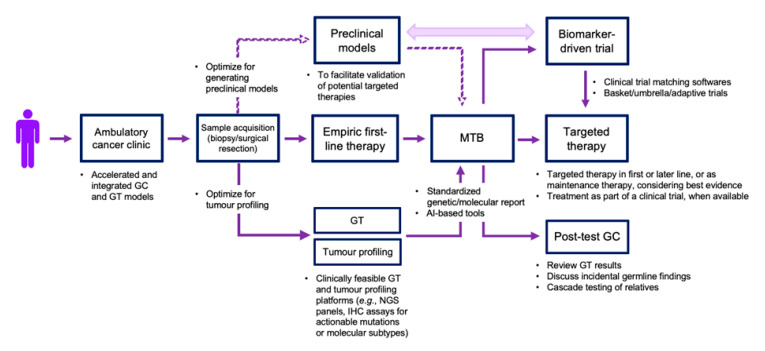
Framework outlining strategies to facilitate implementation of PDAC precision oncology in clinical care. Dotted lines highlight the value of generating preclinical models in parallel, if such platforms are available. Double-headed arrow depicts the potential use of preclinical models to (1) inform precision oncology trial designs and (2) gain mechanistic understanding of treatment responses and acquired treatment resistance observed in biomarker-driven clinical trials. GC, genetic counselling; GT, germline testing; MTB, molecular tumor board; AI, artificial intelligence.

**Table 1 genes-11-01098-t001:** Landmark clinical trials leading to the current empiric chemotherapy regimens for advanced pancreatic ductal adenocarcinoma (PDAC). * Trial design did not include stratification based on *KRAS* or *EGFR* mutational status.

	Year	Investigational Therapy	Comparator Therapy	Overall Survival (Months)
Burris et al. [5]	1997	Gemcitabine	Fluorouracil	5.65 vs. 4.41
Moore et al. [6] NCIC CTG PA.3	2007	Gemcitabine + Erlotinib *	Gemcitabine + placebo	6.24 vs. 5.91
Conroy et al. [3] PRODIGE 4/ACCORD 11	2011	FOLFIRINOX	Gemcitabine	11.1 vs. 6.8
Von Hoff et al. [4] MPACT	2013	Gemcitabine + Nab-Paclitaxel	Gemcitabine	8.5 vs. 6.7
Wang-Gillam et al. [7] NAPOLI-1	2016	Nanoliposomal irinotecan + fluorouracil + folinic acid	Fluorouracil + folinic acid	6.1 vs. 4.2

**Table 2 genes-11-01098-t002:** Landmark clinical trials leading to the current empiric adjuvant chemotherapy regimens for resected PDAC. mFOLFIRINOX, modified FOLFIRINOX.

	Year	Investigational Therapy	Comparator Therapy	Overall Survival (Months)
Neoptolemos et al. [8] ESPAC-1	2004	Fluorouracil	No adjuvant therapy	20.1 vs. 15.5
Oettle et al. [9] CONKO-001	2013	Gemcitabine	No adjuvant therapy	22.8 vs. 20.2
Neoptolemos et al. [10] ESPAC-4	2017	Gemcitabine + Capecitabine	Gemcitabine	28.0 vs. 25.5
Conroy et al. [11] NCIC CTG PA.6	2018	mFOLFIRINOX	Gemcitabine	54.4 vs. 35.0

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
