# Peer review of "Building towards Precision Oncology for Pancreatic Cancer: Real-World Challenges and Opportunities"

_genes, 2020, doi:10.3390/genes11091098_

Round 1

Reviewer 1 Report

The manuscript entitled "Building towards precision oncology for pancreatic cancer: real-world challenges and opportunities" is interesting and avant-garde for the subject matter. The literature supporting the manuscript is adequate as well as the English and the images.

However, I would like to give some suggestions to the authors.

1) Having performed a summary of the genes mutated during pancreatic cancer, it seems only right to mention that during this pathology, Cell-Free DNA Methylation plays a fundamental role.

2) That Cell-Free DNA Methylation can be evidenced by NGS.

For this reason I would invite the authors to review the discussion by expanding with two supporting papers:

Natale, F.; Vivo, M.; Falco, G.; Angrisano, T. Deciphering DNA methylation signatures of pancreatic cancer and pancreatitis. Clin. Epigenet. 2019, 11, 132

Mariarita Brancaccio, Francesco Natale, Geppino Falco, and Tiziana Angrisano. Cell-Free DNA Methylation: The New Frontiers of Pancreatic Cancer Biomarkers’ Discovery. Genes (Basel). 2019 Dec 23;11(1):14. doi: 10.3390/genes11010014.

Author Response

We thank the reviewer for their insightful suggestions. We agree that a discussion regarding the epigenetic dysregulation observed in PDAC is important. Accordingly, we have added section 3.5, that specifically addresses the prevalence and therapeutic implications of epigenetic alterations in PDAC.

Furthermore, as per the reviewer’s suggestions, we have included a paragraph highlighting the value of DNA methylation-based biomarkers for early detection of PDAC. We have expanded the discussion by including the review papers mentioned by the reviewer. These revisions are highlighted on lines 264-280.

Reviewer 2 Report

it is a very interesting and well documented review

Author Response

We thank the reviewer for their positive comments. There are no specific comments to address.